# SMPL: Simulated Industrial Manufacturing and Process Control Learning Environments

**Mohan Zhang**[1,2], **Xiaozhou Wang**[1], **Benjamin Decardi-Nelson**[3], **Song Bo**[3],
**An Zhang**[3], **Jinfeng Liu**[3], **Sile Tao**[1], **Jiayi Cheng**[1], **Xiaohong Liu**[5],
**DengDeng Yu**[4], **Matthew Poon**[1], **Animesh Garg**[2]

[1]Quartic.ai

`mohan, xiaozhou, bill.tao, jerry, matthew@quartic.ai`

[2]University of Toronto

`zhangmo4, garg@cs.toronto.edu`

[3]University of Alberta

`decardin, sbo, azhang1, jinfeng@ualberta.ca`

[4]University of Texas at Arlington

`dengdeng.yu@uta.edu`

[5]Shanghai Jiao Tong University

`xiaohongliu@sjtu.edu.cn`

## Abstract

Traditional biological and pharmaceutical manufacturing plants are controlled by human workers or pre-defined thresholds. Modernized factories have advanced process control algorithms such as model predictive control (MPC). However, there is little exploration of applying deep reinforcement learning to control manufacturing plants. One of the reasons is the lack of high fidelity simulations and standard APIs for benchmarking. To bridge this gap, we develop an easy-to-use library that includes five high-fidelity simulation environments: BeerFMTEnv, ReactorEnv, AtropineEnv, PenSimEnv and mAbEnv, which cover a wide range of manufacturing processes. We build these environments on published dynamics models. Furthermore, we benchmark online and offline, model-based and model-free reinforcement learning algorithms for comparisons of follow-up research. [†]

## 1 Introduction

With a large market value [1], the manufacturing industry is enthusiastic to the ways that is conductive to the production efficiency. A number of studies have demonstrated that reinforcement learning may be applied to manufacturing processes and has the potential to dramatically improve productivity [2, 3].

Our goal is to bridge the gap between deep reinforcement learning research and industrial manufacturing by creating simulation environments that model real-world factories. In this paper, we introduce five manufacturing simulation environments, including beer fermentation, atropine production, penicillin manufacturing, monoclonal antibodies production, and a continuous stirred tank

---

[†]Official documentation:https://smpl-env.github.io/smpl-document/index.html
Official implementation: https://github.com/smpl-env/smpl
Code of experiments: https://github.com/smpl-env/smpl-experiments

simulation. These simulated environments allow us to test the latest advances in reinforcement learning in controlled environments without safety concerns.

In reinforcement learning, the environment is commonly modeled as a Markov Decision Process (MDP). In SMPL, the **state space** is defined by the collection of all reactions, material flows, their concentrations and the state of where the reactions are taking place (e.g. the temperature of a reactor tank). The **initial state** of a trajectory is randomly sampled within a reasonable range, which involves stochasticity. For the **observation space**, there are several levels of observability: there are easily observable states in a real factory that a sensor could essentially measure, for example temperature; there are also some states that are expensive and slow to observe, like some concentrations; there are also some hardly observable states, like the internal changes of some chemical reactions. In the experiments of this paper, we observe everything except the internal changes, forming into the Partially Observable Markov Decision Process (POMDP). To mimic how the actual process is controlled and run, we include 2 categories of actions in the **action space**: 1) manipulable variables. They are the setpoints in the typical control sense (e.g. the temperature of a cooling jacket outside a reactor). 2) input materials. Input materials could be the raw materials that go into the process like sugar, or the attributes of the materials like the concentration. Typically, they can be determined by the operator to optimize the process. The environment **transitions** are modeled with differential equations with first principal or empirical approximations. With the help of domain experts, we managed to access and utilize the industrial data to validate our models and determine the corresponding parameters to align with real-world factories. There are several **reward functions** designed for each of the environments. In the experiments of this paper we only use the dense rewards that measure performance. More specifically, we want a successful control algorithm to be first and foremost safe, then more stable, efficient and productive. Since we have our baseline algorithms, we compare the results of the reinforcement learning algorithms with our baselines. We define more efficient and more productive to be higher in average rewards. We define more stable to be less standard deviation in rewards (which only matters when the algorithms are efficient and productive enough, since a zero-yield function is stable on itself). We would like to clarify that "stable" or "stability" as used throughout this article does not refer to the notion of stable or stability in control theory. Note that in real-life circumstances, efficiency means less product investment (e.g. less sugar and Biomass input as of BeerFMTEnv) and productivity means production yield (e.g. more beer produced as of BeerFMTEnv). But in this paper, our reward function takes both production investment and production yield into account, so we use reward as our single measurement criteria here.

In summary, the main contributions of this paper include: (i) we build five novel manufacturing simulation environments with high fidelity to facilitate researches in these areas; (ii) we tune advanced control algorithms used in industry for the simulation environments, as comparable baselines; (iii) we benchmark popular online and offline, model-based and model-free reinforcement learning algorithms for future reference. In this work, we aim to build simulation environments, and encourage the community to utilize them, in order to find deep reinforcement learning algorithms that solve the real-life environments.

## 2  Related Work

Since the success of Deep Q-Learning (DQN) in Atari games [4], a variety of environments have been developed, including games [5, 6, 7, 8, 9, 10], kinematics [11], autonomous driving [12], recommendation systems [13], multi-agent collaborations [14, 15], networks [16], and offline reinforcement learning data collections [17], which can be used to evaluate deep reinforcement learning algorithms. However, there is a lack of established environments for process control in manufacturing. Considering the vast differences, it is difficult to determine whether a deep reinforcement learning algorithm, which works well in popular benchmarks such as Atari games and dm_control, can be successfully adapted for a production setting owing to the domain discrepancy.

Atari games have discrete image observations and discrete actions, whereas SMPL has continuous observations and actions. Also in Atari games, the visual observations would not have a drastic change in one step. In chemical manufacturing, however, the pH (state) may not respond to a continuous flow of acid (action) after several minutes or even hours, but can also vary greatly with only a slight change in the concentration of such an input like in titration. As opposed to games where one can pause and resume, a factory cannot not wait for the computation to finish before taking the action. The runtime of control algorithms should also be taken into consideration. As compared to other

Advanced Process Control (APC) algorithms, inference of deep reinforcement learning algorithms is faster.

As compared to dm_control (or other MuJoCo [18] based physics simulation environments) which is also continuous in states and actions, our environments are still different. Firstly, each of the states in dm_control has specific and accurate low-level physical semantics, like angle, coordinate, or velocity of a joint. It only records the transition of parts as a kinematical abstraction. Whereas in SMPL, the states are flow rates, the volume of liquid, the concentration of the solution, etc. Since "more is different" [19], SMPL models things at a much larger scale, even though still built with differential equations. In SMPL, those differential equations are modeling physical, chemical and biological processes in manufacturing simulations, and many of them are empirical or phenomenological approximations, so distortions could be a big problem outside their range. To avoid distribution shift between simulations and actual factory transition dynamics, we need to restrict the action and state spaces to a relatively small range. Again, similar to the image represented Atari game states, the position and velocity of a dm_control object would not change suddenly. SMPL, on the other hand, can sometimes change drastically.

To summarize, SMPL is challenging for learning algorithms that work well on Atari games or dm_control simulations. Firstly, SMPL has qualitatively different dynamics. The rate of state change with respect to actions can vary widely. A small change in actions may result in a huge change in states, and the effect of a large change in actions may be delayed for hours. Secondly, as compared to games or robot walking, the punishment of failure is harsh in SMPL, due to security concerns similar to autonomous driving. Thirdly, we observed that the reinforcement learning algorithms tend to exploit the simulators. But due to the complexity of chemical and biomedical reactions, our simulations can easily break or occasionally reach an undefined state when the agent is exploring outside the reliable region. Therefore, compared to the results of reinforcement learning experiments, a fairly simple hand-tuned Proportional–Integral–Derivative (PID) controller can provide tolerable performance. With SMPL, we hope to enable deep reinforcement learning researchers to address this interesting gap.

There have also been efforts in applying reinforcement learning in manufacturing problems [20, 21, 22]. However, to the best of our knowledge, there has not been any other work that provides open-source standardized biochemical process control environments for reinforcement learning and advanced control. More typically, to utilize simulated environments for process control, researchers need to spend sufficient time to understand the underlying mathematical equations, prepare and further develop the environments. SMPL pitched those pain points for reinforcement learning researchers to develop their solutions for manufacturing environments.

## 3    Environments

The SMPL supplements several process control environments to the OpenAI Gym family [5], which alleviates the pain of performing Deep Reinforcement Learning algorithms on them. Furthermore, we provided D4RL-like [17] wrappers for accompanied datasets, making offline reinforcement learning in those environments even smoother. Details of each of the environments can be found in Table 1.

The transitions of the environments are based on Ordinary Differential Equations (ODEs) described in Appendices A. The ODEs themselves model chemical, biological and mechanical transitions and reactions after taking an action from our action space, returning the state and evaluated reward. Originally from a control-theory perspective, we have adapted the transition dynamic to be time-invariant. In real life, the transition contains uncertainty due to the chemical, biological and mechanical process or the sensors' error, but as described in Section 3.6, the noises are hard to model and cannot be simplified as Gaussian noises. The action space and state space are all continuous, with their safety constraints respectively.

### 3.1    ReactorEnv

This Continuous Stirred Tank Reactor (CSTR) process model is a representation of the most common container for a continuous reaction to take place. Even though we have already configured it for a particular reaction, it could easily be re-configured for other tasks (for example, change the cooling jacket to a heating jacket if the reaction is endothermic).

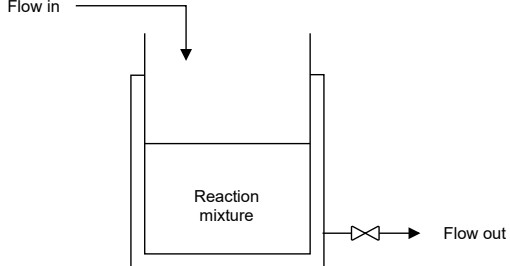

Figure 1: Process flow diagram of the continuous manufacturing process

A schematic diagram of a CSTR is presented in Figure 1. In this article, we consider the case where a single first-order irreversible exothermic reaction of the form A $\rightarrow$ B takes place in the reactor. Because it is a continuous process, the reactants and the products are continuously fed and withdrawn from the reactor, respectively. Because the reaction is exothermic, thermal energy is removed from the reactor through a cooling jacket. The following assumptions are also taken in deriving the model:

- The reaction mixture is well mixed. This implies that there are no spatial variations in the reaction mixture.
- Heat losses to the surroundings are negligible or nonexistent.

The reaction details and the configurations can be found in Appendix A.1.

### 3.2 AtropineEnv

Atropine is a common tropane alkaloid and anticholinergic medication used to treat certain types of nerve agents and pesticide poisonings as well as some types of slow heart rate and to decrease saliva production during surgery. Figure 2 shows the process flow diagram for the atropine production process as presented in the work by Nikolakopoulou, von Andrian and Braatz [23]. This environment simulates a continuous-flow manufacturing process of atropine production . The simulation consists of three tubular reactors ($R_1$, $R_2$, $R_3$) in series and a liquid-liquid separator. Each reactor has a mixer proceeding it where the streams ($S_i$) containing the reactants are thoroughly mixed before being fed into the tubular reactor downstream of it. The end goal is to produce as much atropine as possible while keeping the reactor safe. Details can be found in Appendix A.2.

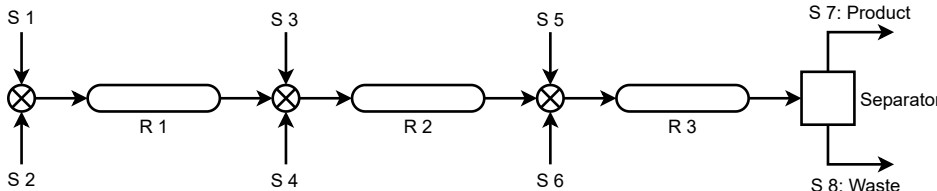

Figure 2: Process flow diagram of the continuous manufacturing process. Details can be found in Table 4

### 3.3 mAbEnv

This environment simulates a manufacturing process of monoclonal antibodies (mAbs), which are widely used for the treatment of autoimmune diseases, cancer, etc. According to a recent publication, mAbs also show promising results in the treatment of COVID-19 [24]. Integrated continuous manufacturing of mAbs represents the state-of-the-art in mAb manufacturing and has attracted a lot of attention, because of the steady-state operations, high volumetric productivity, reduced equipment size and capital cost, etc. However, there is no existing mathematical model of the integrated manufacturing process and there is no optimal control algorithm for the entire integrated process.

This project fills the knowledge gaps by first developing a mathematical model of the integrated continuous manufacturing process of mAbs.

The manufacturing process contains an upstream and a downstream process, and the end goal is to recover as much mAb as possible. Details can be found in Appendix A.3.

### 3.4 PenSimEnv

Penicillin is the first-discovered group of antibiotics in human history. In this environment, we simulate the industrial-scale penicillium chrysogenum fermentation. The aim is to optimize the penicillin production per episode (or batch yield) while avoiding extreme inputs, outputs, or changes that can potentially break the reactor. Figure 3 shows the fermentation process. The simulation environment is based on this industrial-scale fed-batch fermentation simulation [25]. Details can be found in Appendix A.4.

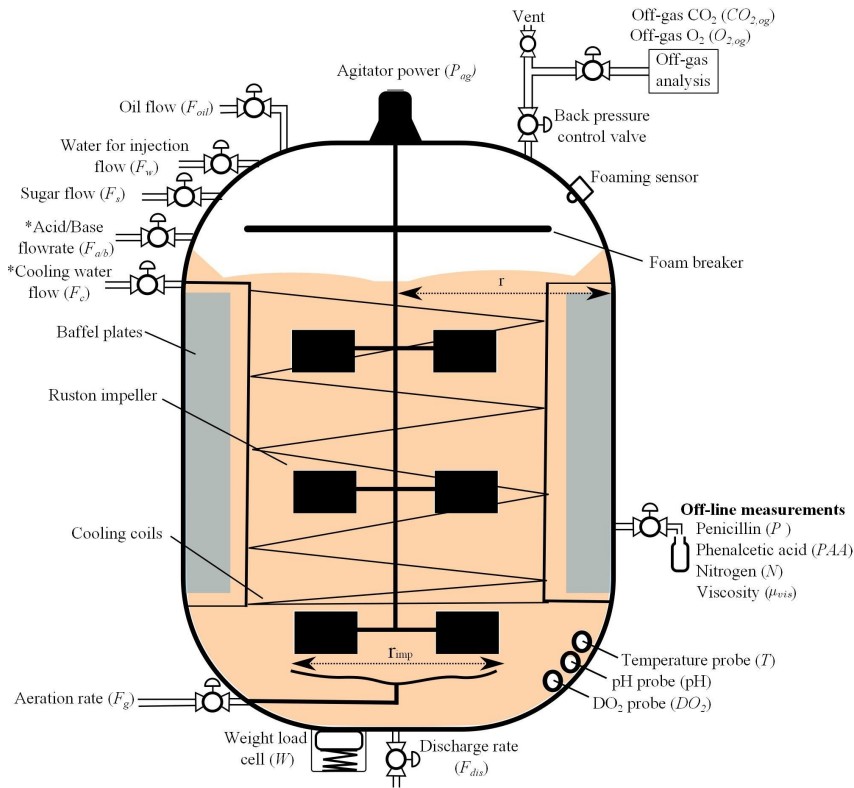

Figure 3: A penicillin manufacturing process, from [25]

### 3.5 BeerFMTEnv

Although there are many different ways of beer production, typically, beer is produced in a fermentation unit where favorable conditions are created for the fermentation of the raw material. Optimal control of the beer fermentation process is a time-optimal control problem, which implies that the goal is to minimize the time required to completely ferment the raw materials. This environment provides a typical and simple enough simulation of the industry-level beer fermentation process. The manipulated variable for this environment is the reaction temperature. As mentioned earlier, the goal of this process is to reach the desired fermentation level within the shortest time possible.

Since we just have a canonical industrial production formula that can only solve from a fixed initial state, we would only perform online reinforcement learning experiments. Details can be found in Appendix A.5.

## 3.6 Limitations

Finally, while enabling machine learning practitioners and control engineers to explore more optimal control strategies and obtain actionable insights for real-world processes, the environments come with some limitations. Firstly, process variability and batch inconsistency is one of the major challenges in manufacturing, which cannot be simply described as gaussian noise. Adding more realistic noise and variability would make the simulations much closer to actual processes. Secondly, full accessibility to states may be difficult. For example, spectrophotometers are usually used for measuring chemical substances like concentrations, which could be costly. Last but not the least, state constraints are not fully considered. For example, due to safety and economic concerns, the next actions may not deviate too much from current actions. Although with the limitations described above, we believe that releasing the environments is still beneficial to researchers interested in applying reinforcement learning to control manufacturing processes.

## 4 Baseline Algorithms

In this section, we compare the performance of our deep reinforcement learning algorithms with a range of different baseline algorithms for each environment. Those baseline algorithms were tuned by experts on the environment. "We tuned the PID, MPC and EMPC by varying either the weights or the horizon, or both until we obtained closed-loop trajectories with little to no overshoot and fast response times. For offline reinforcement learning algorithms, we use the dataset generated by baseline algorithms as the training dataset.

**Bayesian Optimization (BO) [26]**

Bayesian optimization is a sample-efficient optimization method that focuses on solving the black-box optimization problem. In PenSimEnv, we maximize the penicillin yield by collecting 10 random trajectories as our start and then run 1000 Bayesian optimization searches. We use these 1000 trajectories as our offline reinforcement learning training dataset.

**Proportional–Integral–Derivative (PID) Control** PID control is a classical feedback-based control loop that is widely used in industrial control systems. The controller itself is modulated, which computes and tries to minimize the difference between the current state and a set-point (target).

**Advanced Process Control (APC)** APC is a collection of control algorithms that is used in conjunction with PID to further improve the performance of industrial processes. Compared to reinforcement learning algorithms, APCs typically make use of a process model to predict the future evolution of process under the the selected control actions. In this article, we focus on two well known APC algorithms, namely setpoint tracking model predictive control (MPC) and economic model predictive control (EMPC).

**MPC** also known as receding horizon control is an advanced process control algorithm that is able to handle systems with many states and constraints [27]. For this reason, it has had tremendous success in the chemical process industry. MPC requires a mathematical description of the process – either from first principle or empirical – to make predictions about the future evolution of the plant. In MPC, the economic performance objective of the process is translated to minimizing a quadratic cost function which measures the deviation of the system state and input from a desired setpoint. The setpoint is determined and updated by solving a steady-state economic optimization problem in a higher decision making layer known as real-time optimization (RTO). Thus, the economic performance of MPC is as good as the setpoint being tracked and the frequency of the setpoint update.

**EMPC** is a variant of MPC that has gained tremendous attention in the process control community [28]. Compared to MPC, EMPC uses a more general cost function which generally reflects some economic indicator such as waste minimization or yield maximization. From a theoretical point of view, the performance of EMPC is no worse than that of MPC [29]. This is because EMPC simply directly optimizes the process economics compared to MPC which requires that the economic objective be translated to a setpoint tracking objective.

There are still many challenges faced by MPC and its variants.

For example, when the dimension of the system model is too high, or when there are integer control inputs, it might be difficult to solve the MPC optimization problem in real time and within a reasonable

| Env \ Config | Baseline | Traj # | $\mu_r$ | $\sigma_r$ | a_dim | o_dim | max_s | e_r |
|---|---|---|---|---|---|---|---|---|
| ReactorEnv | MPC | 125,000 | -3.7528 | 60.4470 | 2 | 3 | 100 | -1000 |
| AtropineEnv | MPC | 10,000 | -20.6094 | 339.5105 | 4 | 39 | 60 | -100000 |
| PenSimEnv | BO | 1000 | 3.3071 | 1.4673 | 6 | 9 | 1150 | -100 |
| mAbEnv | MPC | 1000 | 1322.5012 | 174.7515 | 9 | 1970 | 200 | -100 |
| BeerFMTEnv | N/A | N/A | N/A | N/A | 1 | 8 | 200 | -200 |

Table 1: Env and Dataset Details. For each environment, from left to right: "Baseline" is the name of the baseline algorithm that the dataset is generated with; "Traj #" is the number of trajectories; "$\mu_r$" is the mean of the rewards of the dataset; "$\sigma_r$ is the standard deviation of the rewards of the dataset; "a_dim" (represents "action_dim") is the dimension of action space in the environment; "o_dim" (represents "observation_dim") is the dimension of observation space in the environment; "max_s" (represents "max_steps") is the maximum number of steps possible in the environment; "e_r" is called "error_reward", which is given to a failed trajectory in the environment. Note that error_reward always satisfies $error\_reward \le r_{min} \times max\_steps$, where $r_{min}$ is the least possible reward for a step in the environment.

time. EMPC faces a significantly higher computational challenges than MPC. This is because a more general dynamic economic optimization need to be solved in real time. Moreover, handling uncertainties in MPC or EMPC is a very challenging task [30]. DRL on the other hand, works well under uncertainty with fast inference time.

## 5  Experiments

In this section, we demonstrate the results of our experiments. In the experiments, each offline reinforcement learning algorithm is trained for 500 epochs, and each online reinforcement learning algorithm is trained for 2 million environment steps. These experiments can be treated as a benchmarking baseline for future research in achieving higher stability, efficiency and productivity. For the environments, we mimic the design of the OpenAI Gym so that we can easily train and test a variety of reinforcement learning algorithms on them. The associated code can be accessed with the provided code.

### 5.1  Offline Reinforcement Learning

The simulations are meant to mimic and abstract existing factories, and the traditional manufacturing plants are controlled by human workers or threshold functions. Luckily, most of the states and actions are recorded by sensors and workers. A natural thought would be to use those historical data to learn better control algorithms, and a successful control algorithm should be more stable, efficient and productive. Offline reinforcement learning is a possible approach since it can purely learn from historical data. As in real life, it is infeasible to train RL agents on actual manufacturing production lines due to safety and economic concerns. With offline reinforcement learning, a control policy can be developed by learning from offline data collected from manual control or APC.

Table 2 shows the experiment results. Each of the four simulation environments has an expert control algorithm which is manually tuned by experts. This expert control algorithm can provide successful controls in non-extreme cases, with potentially low efficiency due to online optimization and computation complexity.

We set a maximum time limit for each of the environments. If a trajectory generated by an algorithm in an environment ends before the maximum time limit due to an error reported by the environment, or if an action/observation goes beyond the allowed limit, then this trajectory is marked as failed; otherwise, this trajectory is marked as succeeded. A failed trajectory receives a negatively large reward as the punishment (the name of which is error_reward). Since the initial states are not fixed, a successful algorithm needs to adapt to a wide range of situations.

Specifically as Table 1 shows, ReactorEnv, AtropineEnv and mAbEnv all have MPC as their baseline algorithm. PenSimEnv uses a version of Gaussian Process based Bayesian Optimization (GPEI)

Table 2: Offline and Online Experiment Results Part 1

| Env / Algo | ReactorEnv | | AtropineEnv | | PenSimEnv | | mAbEnv | |
|---|---|---|---|---|---|---|---|---|
| OfflineRL | $\mu_r$ | $\sigma_r$ | $\mu_r$ | $\sigma_r$ | $\mu_r$ | $\sigma_r$ | $\mu_r$ | $\sigma_r$ |
| *MPC* | *-0.1912* | *15.9328* | *-20.6094* | *339.5105* | *N/A* | *N/A* | *1322.5012* | *174.7515* |
| *GPEI* | N/A | N/A | N/A | N/A | *3.3071* | *1.4673* | N/A | N/A |
| EMPC | N/A | N/A | N/A | N/A | N/A | N/A | 1314.1145 | 221.4624 |
| PID | -1.0823 | 35.0924 | N/A | N/A | N/A | N/A | N/A | N/A |
| PLAS [31] | -0.1909 | 27.9975 | **-4.8416** | 6.7930 | 2.0421 | 4.7936 | 1324.1982 | 124.7369 |
| PLASWithPert [31] | -0.1638 | 13.2464 | -17.5712 | 6.1863 | **3.0056** | 4.7353 | 1357.0842 | 122.1667 |
| TD3 [32] | -0.6428 | 25.4631 | -51.3783 | **4.1409** | -1.2575 | 11.1618 | 708.1167 | 571.8498 |
| AWAC [33] | **-0.0965** | **13.9237** | -8.4070 | 10.8725 | 2.3288 | 4.6710 | **1374.2244** | **61.2931** |
| BEAR [34] | -13.4975 | 111.0537 | -28.2412 | 11.6400 | 2.2086 | **4.4271** | 1268.8020 | 297.5198 |
| BCQ [35] | -2.0008 | 43.3134 | -18.2633 | 4.6889 | 2.1341 | 4.6710 | 1247.0172 | 313.2412 |
| SAC [36] | -0.7120 | 18.5272 | -181.0055 | 5.4775 | 2.5961 | 4.8317 | 1197.1507 | 503.1202 |
| DDPG [37] | -1.3219 | 62.6639 | -203.0525 | 5.7211 | -1.9732 | 13.9105 | 711.1741 | 574.0687 |
| CQL [38] | -2.5266 | 79.5985 | -13.9241 | 8.9072 | 2.5543 | 4.5791 | 1254.5728 | 233.4812 |
| COMBO [39] | -7.9475 | 84.2047 | -30.1022 | 11.2495 | 3.1368 | 4.5718 | 1165.3812 | 440.0366 |
| MOPO [40] | -8.8802 | 94.0311 | -90.8561 | 52.7119 | 1.4345 | 5.0138 | -100.0000 | 0.0000 |
| BC [41] | -3.5765 | 59.8082 | -5.4473 | 44.1707 | 0.7440 | 4.5213 | 1322.8184 | 111.1911 |
| OnlineRL | | | | | | | | |
| PPO [42] | -31.9667 | 180.9413 | -67.4412 | 2595.8698 | 2.5231 | 4.6745 | -92.7862 | 137.6964 |
| A3C [43] | -1000.0000 | 0.0000 | -50.2008 | 2239.2415 | -0.8551 | 9.3091 | -88.8452 | 272.8659 |
| ARS [44] | -1000.0000 | 0.0000 | -83333.3879 | 37267.6776 | | | -94.3842 | 184.2320 |
| IMPALA [45] | -1000.0000 | 0.0000 | -100000.0000 | 0.0000 | -2.0575 | 14.1869 | -94.3842 | 184.2320 |
| PG [46] | -140.8971 | 311.5265 | -102.5358 | 3194.3687 | 2.1039 | 4.3759 | -86.1046 | 241.7652 |
| SAC | -113.3525 | 293.3941 | -16.4201 | 1280.6835 | -2.3044 | 15.0070 | -90.1491 | 233.4410 |
| DDPG | -97.6231 | 279.8284 | -141.0357 | 302.8068 | -0.8373 | 9.1140 | -95.1542 | 288.9491 |

| BeerFMTEnv / OnlineRL | $\mu_r$ | $\sigma_r$ |
|---|---|---|
| PPO | 1.0688 | 20.7031 |
| A3C | **0.8706** | 20.3378 |
| ARS | -200.0000 | 0.0000 |
| IMPALA | -1.0371 | 14.5516 |
| PG | -1.9900 | **14.0014** |
| SAC | -1.9753 | 14.0415 |
| DDPG | -2.1842 | 15.3056 |

developed in-house. We generate datasets with these algorithms, to train our reinforcement learning algorithms offline. Furthermore, ReactorEnv has a PID controller and mAbEnv has an EMPC controller for comparison. We did not perform offline reinforcement learning on the BeerFMTEnv since it only has a static rule as the baseline. The amount of data sampled differently because of the cost of sampling. The environments are all drastically different from each other and cover different types of manufacturing processes.

D4RL [17] is one of the most widely used benchmarks in the offline reinforcement learning community. Thus, we make the generation of the dataset in both D4RL and Torch format with any control algorithm possible in our library, and only a few lines of code would be sufficient.

Thanks to D3RLPY [47], we can perform batched parallel training with a little engineering. The results are shown in Table 2:

On ReactorEnv, only Advantage Weighted Actor-Critic (AWAC) is able to outperform the MPC baseline. In terms of average rewards, Policy in the Latent Action Space (PLAS) and PLAS with perturbation slightly go beyond our baseline.

On AtropineEnv, PLAS has the maximum average rewards, while Twin Delayed Deep Deterministic Policy Gradients (TD3) has the smallest standard deviation of rewards. Moreover, AWAC, Batch-Constrained Q-learning (BCQ), Conservative Q-Learning (CQL) and Behavior Cloning (BC) are able to provide performances beyond our Model Predictive Control (MPC) baseline. Among them, we could say that PLAS, BC and AWAC solved the AtropineEnv better than the MPC baseline for they have a 50% more average and a 50% less standard deviation of rewards.

On PenSimEnv, it is also PLAS that has the highest average rewards, while Bootstrapping Error Accumulation Reduction (BEAR) has the lowest standard deviation of rewards. However, none of the offline reinforcement learning algorithms can produce average rewards higher than the Gaussian Process based Bayesian Optimization (GPEI) baseline.

On mAbEnv, AWAC wins the highest maximum average and the smallest standard deviation of rewards. PLAS with perturbation, PLAS and BC all have slightly better performances compared to our baseline MPC in terms of average and standard deviation.

PLAS [31] seems to achieve a better performance than the baseline in many environments. Moreover, on mAbEnv, PLAS with perturbation is better, because we only have 1000 MPC trajectories. For AtropineEnv, PLAS without perturbation has higher performance, since AtropineEnv has 10,000 trajectories. On ReactorEnv, PLAS with our without perturbation has a very similar performance, and there are 125,000 trajectories. From these experiments, we observe that the perturbation layer, which aims to sample generalized action out of the training dataset, could harm the performance when the training dataset is too small, but can improve the performance when the dataset is large enough.

AWAC, with its ability to utilize sub-optimal data-points based on traditional advantage actor-critic algorithms [43], is able to thrive in both data-abundant (ReactorEnv) and data-scarce (mAbEnv) scenarios. The fact that it outperforms all other offline reinforcement learning algorithms makes its data-utilization trick worthwhile for further exploration.

Deep Deterministic Policy Gradients (DDPG) has poor performance in most environments, as compared to other q-learning variants, even when provided dense rewards [48]. Some researches shows that without careful tuning of the hyperparameters, DDPG and Soft Actor-Critic (SAC) tend to largely overestimate the q-value during the training, and the exploding imitation value estimation leads to an inescapable extrapolation error [35]. However, when exploration is allowed, DDPG could have a slightly better performance. Compared to DDPG and SAC, which were not designed for offline reinforcement learning, BCQ claims to be able to improve on restricting the action spaces, but it still suffers from the q-value overestimation in PenSimEnv and mAbEnv.

There are some algorithms (like A3C, ARS, IMPALA in ReactorEnv and IMPALA in AtropineEnv) that always tend to break the simulation by either going out of the limit or reaching an unacceptable state. With careful tuning, their performance could be improved.

## 5.2 Online Reinforcement Learning

In real-life use cases, we cannot directly apply q-learning or policy gradient on the producing plant for a hundred thousand episodes to train a good RL agent. However, we have the simulations anyways, we would like to perform online reinforcement learning experiments on those simulations, and the results can serve as baselines.

We utilize the Ray [49] library, with its RLlib [50] and Tune [51] components to parallelize the training. With a little tuning and a limited budget of training time, the performance of online reinforcement learning algorithms is all lower than the offline reinforcement learning algorithms. We believe that the exploration in SMPL is hard in general, and the guidance provided by expert algorithms can provide successful controls in most cases (with potentially low efficiency).

Note that compared to the offline reinforcement learning experiments, SAC shows worse results. DDPG, on the other hand, is able to show a slightly better result in PenSimEnv. The hypothesis could be that, due to the very slight change in rewards (the dataset of PenSimEnv has a very small standard deviation of rewards, only 1.4673), the over-estimation problem of DDPG might be fixed by exhaustive exploration.

# 6 Conclusions and Future Work

In our work, we introduced five simulation environments that covered a wide range of manufacturing processes. Corresponding with the environments, we provided expert-tuned control algorithms that are employed in factories. Based on the built environments and baselines, we tried offline and online, model-based and model-free reinforcement learning algorithms.

From the experiment results, we suggest utilizing offline reinforcement learning algorithms and learning from the baseline-generated samples as a starting point.

Only on AtropineEnv, a few reinforcement learning algorithms show a significant improvement as compared to baselines, while on other environments the reinforcement learning approaches are only marginally better, if at all. Therefore, targeted research, specifically designed or carefully tuned algorithms might be a prerequisite to succeed on our environments, for 1. our environments are challenging and complex (especially mAbEnv, PenSimEnv and ReactorEnv) 2. there are significant differences between our environments and the environments where the reinforcement learning algorithms are originally designed or often experimented.

We will continue to work on the existing environments to find a more stable, efficient and productive algorithm. Experiments already show us that directly applying existing algorithms might not be a solution, so further research on how to sample safe actions from empirical results given only historical data might be a good starting point. The overall good performance of PLAS hints that digging into shaping the latent action space might be a good idea.

Another one of our goals is to find an algorithm that can perform well in all the environments with little modification, which can be useful if a plant decides to change a reactor tank, or we want to adopt a trained algorithm for a new plant. We would like to develop existing meta-learning algorithms like [52, 53].

In addition to the existing configurations, we would also like to add more tunable parameters to the environments that can represent different types of manufacturing processes. More simulation environments, on top of the existing five, would be added to the family once finished.

We would actively develop and maintain this library, to better serve the reinforcement learning community.

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
