# OpenReview forum: "SMPL: Simulated Industrial Manufacturing and Process Control Learning Environments"
_NeurIPS.cc/2022/Track/Datasets_and_Benchmarks — NeurIPS 2022 Datasets and Benchmarks _

### Official Review · Reviewer_ypAC · 2022-07-26
**Some very valuable environments**

**Rating:** 7
**Confidence:** 3
**Correctness:** The paper correctly states its contri…
**Clarity:** The paper is written clearly and is w…

**Strengths:**

1.  A very good entry point: a simulation environment for industrial applications of reinforcement learning
2. The simulation environment is based on published dynamics models, which enhances the reliability of the environment
3. Wide range of baseline experiments available

**Weaknesses:**

1. Unclear about the difficulty or cost of modifying or adding new environments if others wish to do so
2. It is not clear why these environments were chosen? Do they have complementary qualities or is there another reason?

**Additional Feedback:**

None

**Documentation:**

The author needs to add further

**Relation To Prior Work:**

Not very clear, the authors need to have more discussion.

**Summary And Contributions:**

The authors developed an easy-to-use library based on published dynamics models on five high-fidelity simulation environments: BeerFMTEnv, ReactorEnv, AtropineEnv, PenSimEnv, and mAbEnv, which cover a wide range of production processes. and were tested with different algorithms. This is of significant help for industrial applications of reinforcement learning.

---

> ### Author Response · Authors · 2022-08-13
> **Thanks for your feedback and we've revised the paper! To address the weaknesses and concerns…**
>
> **Weaknesses:**
>
> > 1. Unclear about the difficulty or cost of modifying or adding new environments if others wish to do so
>
> Thank you for the comment.
> We aim to relieve the pain of customizing or adding new environments, by providing clear documentation here https://smpl-env.github.io/smpl-document/index.html.
> If a researcher wants to add or customize an environment, the person needs to follow the design of smplEnvBase in https://github.com/smpl-env/smpl/blob/main/smpl/envs/utils.py.
> The difficulty of customizing or adding new environments depends on the change being made and the complexity of the environments themselves. However, the authors would happily and willingly aid the changes or any inquiries, with GitHub issues or personal contacts.
> The SMPL package is under active development and any contributions, comments or suggestions are very welcome!
>
> > 2. It is not clear why these environments were chosen? Do they have complementary qualities or is there another reason?
>
> Thank you for the comment. While all of them are biochemical processes, we tried to cover different types. For example, we have batch process (e.g. PenSimEnv) and continuous process (e.g. AtropineEnv), single unit process (e.g. BeerFMTEnv) and multi-unit process (e.g. mAbEnv).
>
> **Relation To Prior Work:**
> > Not very clear, the authors need to have more discussion.
>
> Thank you for your comment. We have revised our paper to include related works on applying RL to manufacturing problems.
> “There have also been efforts in applying reinforcement learning in manufacturing problems (https://www.sciencedirect.com/science/article/abs/pii/S0166361520306072, https://dl.acm.org/doi/10.1145/3424311.3424326, https://link.springer.com/article/10.1007/s44163-021-00003-3). However, to the best of our knowledge, there has not been any other work that provides open-source standardized biochemical process control environments for reinforcement learning and advanced control. More typically, to utilize simulated environments for process control, researchers need to spend sufficient time to understand the underlying mathematical equations, prepare and further develop the environments. SMPL pitched those pain points for reinforcement learning researchers to develop their solutions for manufacturing environments.”
>
> **Documentation:**
>
> > The author needs to add further
>
> Thank you for your feedback. Based on other reviewers’ comments, we have added more contents, including more related works, detailed discussion of limitations, better description of baseline algorithms, the MDP processes of the environments and the experiment outcomes. Due to the space limitation, there might still be confusion in the paper. Do you mind to help us by pointing out where you would like further improvements?

---

### Official Review · Reviewer_dVtT · 2022-07-27
**Simulation package for evaluating reinforcement learning for manufacturing processes**

**Rating:** 7
**Confidence:** 4

**Strengths:**

1. The Simulated Industrial Manufacturing and Process Control Learning Environments (SMPL) software with several process control environments is a plus.

2. There is a good amount of detail on the different manufacturing pipelines in the supplement.

3. There is an extensive list of offline and online RL algorithms being evaluated.


**Weaknesses:**

1. It is not clear how broadly useful the simulation package will be. The similarities and the differences between the five manufacturing pipelines should have been brought out better. This is important for a potential adopter to decide whether and how to adopt this package for her environment.

2. The software package seems to be incomplete compared to what is presented in the paper. I could find only one model (atropine_process) and three datasets instead of the five that are mentioned in the paper and in the documentation.

3. There is too little detail to appreciate why the presented baselines would qualify as baseline algorithms. For example, the paper says (lines 159-160) "Those baseline algorithms were tuned by experts on the environment." How?

**Additional Feedback:**

None

**Clarity:**

Reasonably but not completely. There are a few abrupt sentences without justification, which hamper understanding of the claims.
* "This expert control algorithm can provide successful controls in none extreme cases (with potentially low efficiency)." Why would tuning by experts lead to low efficiency?
* "The amount of data sampled differently because of the cost of sampling. The environments are all drastically different from each other and cover different types of manufacturing processes."


**Correctness:**

The software package seems to be well organized. The documentation is of a high quality (https://smpl-env.readthedocs.io/en/latest/index.html) and is much higher standard than typical academic documentation seen in submissions to this forum.

On the weakness side, the way the RL algorithms are evaluated is not completely spelled out. For example, what are the action spaces and rewards, how are the training samples generated, how is it determined if a run is not going to terminate.


**Documentation:**

Yes the software packaging and documentation are praiseworthy.

**Relation To Prior Work:**

I am not familiar with simulation environments for manufacturing processes.

**Summary And Contributions:**

The paper presents a simulation environment, called SMPL, with models for five manufacturing environments. It uses this simulation model to evaluate several offline and online reinforcement learning algorithms. It compares their performance to baseline methods which are tuned by experts.

---

> ### Author Response · Authors · 2022-08-13
> **Thanks for your feedback and we've revised the paper! To address the concerns… (1/2)**
>
> **Weaknesses:**
>
> > It is not clear how broadly useful the simulation package will be. The similarities and the differences between the five manufacturing pipelines should have been brought out better. This is important for a potential adopter to decide whether and how to adopt this package for her environment.
>
> Thank you for the comment. While all of them are biochemical processes, we tried to cover different types. For example, we have batch process (e.g. PenSimEnv) and continuous process (e.g. AtropineEnv), single unit process (e.g. BeerFMTEnv) and multi-unit process (e.g. mAbEnv). Potential users would be able to adopt the environments for their purposes by reading the Appendices (which contain the theoretical details and the ODEs about the dynamics), the documentation https://smpl-env.github.io/smpl-document/api/smpl.envs.html (which contains the detailed APIs) and the code https://github.com/smpl-env/smpl/tree/main/smpl/envs (if they are willing to know the implementation details). They can also always reach out to the authors for collaboration through GitHub issues or personal contacts.
>
> > The software package seems to be incomplete compared to what is presented in the paper. I could find only one model (atropine_process) and three datasets instead of the five that are mentioned in the paper and in the documentation.
>
> Thank you for the comment. The 3 “datasets” are actually the configuration files for the environments. We have renamed the folder to “configdata” for better clarity here https://github.com/smpl-env/smpl/tree/main/smpl/configdata. To generate offline RL datasets, we may refer to “offline_data_generation.py” scripts in https://github.com/smpl-env/smpl-experiments.
>
> > There is too little detail to appreciate why the presented baselines would qualify as baseline algorithms. For example, the paper says (lines 159-160) "Those baseline algorithms were tuned by experts on the environment." How?
>
> Thank you for the comment. Indeed, controller tuning is not an easy task. The MPC and PID baselines were tuned following standard tuning practices. The tuning parameters are the PID gains, MPC weights on the state and inputs, and the prediction horizon. Specifically, we varied the controller tuning parameters until we obtained closed-loop trajectories with little to no overshoot and fast response times. To be more specific, we consult the transition dynamics (ODEs) of the environments while tuning the MPC, the PID and the BO.
> In light of this comment, we have added a statement in the article to briefly clarify how the parameters were tuned: “We tuned the PID, MPC and EMPC by varying either the weights or the horizon, or both until we obtained closed-loop trajectories with little to no overshoot and fast response times.”
>
> **Correctness:**
>
> > On the weakness side, the way the RL algorithms are evaluated is not completely spelled out. For example, what are the action spaces and rewards, how are the training samples generated, how is it determined if a run is not going to terminate.
>
> Thank you for the comment. For better clarity, we have updated the introduction section where action space is introduced, with “To mimic how the actual process is controlled and run, we include 2 categories of actions in the action space: 1) manipulable variables. They are the setpoints in the typical control sense (e.g. the temperature of a cooling jacket outside a reactor). 2) input materials. Input materials could be the raw materials that go into the process like sugar, or the attributes of the materials like the concentration. Typically, they can be determined by the operator to optimize the process.”
> A run will always terminate,  by either 1) reaching the max time steps, which is the time limitation for a run. 2) generating an error / stop signal due to security reasons. For the latter case, an error_reward was given.
> There are multiple configurations for a reward function. For example, in the atropine environment, the environment has an option standard_reward_style, which was documented in https://github.com/smpl-env/smpl/blob/main/smpl/envs/mabenv.py as “ standard_reward_style (str, optional): Reward style, can be 'setpoint', 'productivity' or 'yield'. The 'setpoint' reward bases on how the controller is able to move the observation close to the steady state observation; the 'productivity' reward bases on the MAb upstream productivity; the 'yield' computes the collected mAb yield from downstream. Defaults to 'setpoint'.” All the experiments in our paper used default configuration but there are many configurations provided for future works. Due to the length limitation, we did not include the detailed explanation in the main paper. More details can be found in Appendix A, https://smpl-env.github.io/smpl-document/index.html and https://github.com/smpl-env/smpl/tree/main/smpl/envs

---

> > ### Author Response · Authors · 2022-08-13
> > **Continue… (2/2)**
> >
> > **Clarity:**
> >
> > > Reasonably but not completely. There are a few abrupt sentences without justification, which hamper understanding of the claims.
> > "This expert control algorithm can provide successful controls in none extreme cases (with potentially low efficiency)." Why would tuning by experts lead to low efficiency?
> >
> > Thank you for pointing it out. For better clarity, we have changed the sentence to “This expert control algorithm can provide successful controls in non-extreme cases, with potentially low efficiency due to online optimization and computation complexity.” Namely, the low efficiency is due to the limitation of the used algorithms themselves (e.g. MPC) as compared to the reinforcement learning algorithms. MPC and its variants are much slower than the RL algorithms during inference.
> >
> > > "The amount of data sampled differently because of the cost of sampling. The environments are all drastically different from each other and cover different types of manufacturing processes."
> >
> > Thank you for the comment. We provided a reasonably good amount of data that compute time would allow. Namely, the complexity of each environment affects the sampling time of each trajectory. For example, on an i9-12900k + RTX3090 machine, it took less than two days to sample 125000 trajectories of MPC on ReactorEnv, but it can take three weeks to sample 1000 trajectories of MPC on mAbEnv.

---

### Official Review · Reviewer_RKJs · 2022-07-27
**Review of SMPL - The paper is not following the track's instructions**

**Rating:** 4
**Confidence:** 4
**Correctness:** Yes
**Clarity:** No

**Strengths:**


** The proposal (creation) of five environments with different manufacturing domains is the strongest contribution.


** The paper is well written, and the benchmarking of offline and online RL algorithms are interesting and providing wrappers for accompanied datasets is useful


** The research seems promising for future research applying reinforcement learning to enhance manufacturing domain

**Weaknesses:**


** The paper is violating the datasets and benchmarks track requirements by adding Appendix towards end of main paper, not providing a Checklist, and not providing explanation for the datasets and their usage. The authors need to read carefully the instructions for submitting for such a track.


** The action space is not convincing from my understanding of RL actions in the statement "The action space consists of manipulable variables (e.g., the temperature of a cooling jacket outside a reactor), input materials (sugar for example), and the concentration of input materials when possible."


** In Related Work Section, this comparison of two different applications (Arari and SMPL) does not make sense. Comparing to manufacturing works that use RL makes more sense. For example, the work "Optimization of global production scheduling with deep reinforcement learning [Waschneck et. al. 2018]".



** Why only 3 datasets are given in the GitHub folder for your five tasks?


** Some rewards' numbers are strange (e.g., A3C and ARS) in Table 2. Is there explanation for that?


** Convergence analysis of Reinforcement Learning is missing and the objective itself is missing in the main paper which makes the main point not clear


** Environments Section has many details that may be reduced given the target conference


** No advanced machine learning algorithms as baselines which make the evaluation much weaker


** The authors state "utilize the Ray [46] library, with its RLlib [47] and Tune [48] components to parallelize the training. With a little tuning and a limited budget of training time, the performance of online reinforcement learning algorithms is all lower than the offline reinforcement learning algorithms". Give quantitative numbers for such metrics from your evaluation.


** No clear documentation for the library and usage of data, given the GitHub link.


** The code link (URL) is missing from the main paper which is not helping the reader.


**Additional Feedback:**

See weaknesses for my main concerns

**Documentation:**

No

**Relation To Prior Work:**

Yes, but with missing related work in deep reinforcement learning for manufacturing processes

**Summary And Contributions:**

To apply deep reinforcement learning to control manufacturing processes, the authors introduce an easy-to-use library that includes five simulation environments that cover a wide range of manufacturing processes. The paper also benchmarks several online and offline, model-based and model-free reinforcement learning algorithms for comparison. They are providing wrappers for accompanied datasets.

---

> ### Author Response · Authors · 2022-08-13
> **Thanks for your feedback and we've revised the paper! To address the concerns… (1/2)**
>
> **Weaknesses:**
>
> > 1. The paper is violating the datasets and benchmarks track requirements by adding Appendix towards end of main paper, not providing a Checklist, and not providing explanation for the datasets and their usage. The authors need to read carefully the instructions for submitting for such a track.
>
> Thanks for pointing this out. We have provided the checklist in the latest revision on page 15. Note that the 3 “datasets” are actually the configuration files for the environments. We have renamed the folder to “configdata” for better clarity here https://github.com/smpl-env/smpl/tree/main/smpl/configdata. To generate offline RL datasets, we may refer to “offline_data_generation.py” scripts in https://github.com/smpl-env/smpl-experiments.
>
>
> > 2. The action space is not convincing from my understanding of RL actions in the statement "The action space consists of manipulable variables (e.g., the temperature of a cooling jacket outside a reactor), input materials (sugar for example), and the concentration of input materials when possible."
>
> Thank you for the comment. We provide an action space for a reinforcement learning agent to interact with the environment: In our manufacturing environments, the action space consists of manipulable variables. We choose the actions based on the following two principles: 1. The action should be changeable through an operation, e.g., increasing the temperature of a cooling jacket;  2. The action should have an effect on the environment itself (as the temperature of a cooling jacket could affect the product yield). The action space is made up of those actions.
> We also updated the introduction section where action space is introduced, with “To mimic how the actual process is controlled and run, we include 2 categories of actions in the action space: 1) manipulable variables. They are the setpoints in the typical control sense (e.g. the temperature of a cooling jacket outside a reactor). 2) input materials. Input materials could be the raw materials that go into the process like sugar, or the attributes of the materials like the concentration. Typically, they can be determined by the operator to optimize the process.”
>
>
> > 3. In Related Work Section, this comparison of two different applications (Arari and SMPL) does not make sense. Comparing to manufacturing works that use RL makes more sense. For example, the work "Optimization of global production scheduling with deep reinforcement learning [Waschneck et. al. 2018]".
>
> Thank you for your comment. We have revised our paper to include related works on applying RL to manufacturing problems.
> “There have also been efforts in applying reinforcement learning in manufacturing problems (https://www.sciencedirect.com/science/article/abs/pii/S0166361520306072, https://dl.acm.org/doi/10.1145/3424311.3424326, https://link.springer.com/article/10.1007/s44163-021-00003-3). However, to the best of our knowledge, there has not been any other work that provides open-source standardized biochemical process control environments for reinforcement learning and advanced control. More typically, to utilize simulated environments for process control, researchers need to spend sufficient time to understand the underlying mathematical equations, prepare and further develop the environments. SMPL pitched those pain points for reinforcement learning researchers to develop their solutions for manufacturing environments.”
>
>
> > 4. Why only 3 datasets are given in the GitHub folder for your five tasks?
>
> Thank you for the comment. The 3 “datasets” are actually the configuration files for the environments. We have renamed the folder to “configdata” for better clarity here https://github.com/smpl-env/smpl/tree/main/smpl/configdata. To generate offline RL datasets, we may refer to “offline_data_generation.py” scripts in https://github.com/smpl-env/smpl-experiments.
>
>
> > 5. Some rewards' numbers are strange (e.g., A3C and ARS) in Table 2. Is there explanation for that?
>
> Thanks for pointing that out. The unusual reward numbers are the corresponding error_reward (“e_r” in Table 1). “There are some algorithms (like A3C, ARS, IMPALA in ReactorEnv and IMPALA in AtropineEnv) that always tend to break the simulation by either going out of the limit or reaching an unacceptable state. With careful tuning, their performance could be improved.” (lines 307-309). Therefore, the average reward equals the error_reward, which in ReactorEnv is -1000 and in AtrpoineEnv is -100000, as Table 1 shows.

---

> > ### Author Response · Authors · 2022-08-13
> > **Continue… (2/2)**
> >
> > > 6. Convergence analysis of Reinforcement Learning is missing and the objective itself is missing in the main paper which makes the main point not clear.
> >
> > Thank you for the comment. The reinforcement learning algorithms are neither tuned nor optimized, they are merely provided as baselines for future references. The objective is to develop a (reinforcement learning) algorithm that can be stable, efficient, and productive. Since stability, efficiency and productivity are considered in the reward function (as described in lines 40 to 52), the objective is to maximize the reward.
> >
> > > 7.  Environments Section has many details that may be reduced given the target conference.
> >
> > Thank you for the feedback. We’ve gotten mixed reviews on this. On one hand, it is obviously helpful to understand the MDPs and transition dynamics; on the other hand, it is quite difficult to follow if the readers are not from the process control domain. With that in mind,  we tried to balance both sides. It is much appreciated if you could point out some of the details that you think should be reduced and we will address them.
> >
> >
> > > 8.  No advanced machine learning algorithms as baselines which make the evaluation much weaker.
> >
> > Thank you for your comment. The reinforcement learning algorithms in the paper are not tuned for performance, but provided as baselines for future reference. For that reason, we have covered the commonly used model-based and model-free, online and offline reinforcement learning algorithms, as well as control-based algorithms. Please suggest the advanced machine learning algorithms that you have in mind and we will include them if time permits, or include them in our GitHub repo in the future.
> >
> >
> > > 9. The authors state "utilize the Ray [46] library, with its RLlib [47] and Tune [48] components to parallelize the training. With a little tuning and a limited budget of training time, the performance of online reinforcement learning algorithms is all lower than the offline reinforcement learning algorithms". Give quantitative numbers for such metrics from your evaluation.
> >
> > Thank you for the feedback. For the training metrics,   as stated in line 233 to 235,  “In the experiments, each offline reinforcement learning algorithm is trained for 500 epochs, and each online reinforcement learning algorithm is trained for 2 million environment steps.” The hyperparameters that we used are all in the offline_experiments.yaml and online_experiments.yaml for each of the five environments in https://github.com/smpl-env/smpl-experiments. In fact, most hyperparameters are from the papers as default. “the performance of online reinforcement learning algorithms is all lower than the offline reinforcement learning algorithms”, as Table 2 shows, and the explanation for this result is explained in section 5.2.
> >
> >
> > > 10. No clear documentation for the library and usage of data, given the GitHub link.
> >
> > Thank you for the feedback. As discussed in point 4, the intent of our work is to provide simulation environments, not datasets. Please find the documentation for other parts below:
> > The documentation for the five environments can be found here: https://smpl-env.readthedocs.io/en/latest/index.html;
> > The offline_data_generation.py scripts in https://github.com/smpl-env/smpl-experiments can be used to generate offline RL datasets;
> > Readme for environment configurations: https://github.com/smpl-env/smpl/tree/main/smpl/configdata.
> >
> >
> > > 11. The code link (URL) is missing from the main paper which is not helping the reader.
> >
> > Thanks for pointing this out! We’ve added the links to the paper as the footnote of the first page.

---

### Official Review · Reviewer_FUoV · 2022-07-28
**Review of SMPL: Simulated Industrial Manufacturing and Process Control Learning Environments**

**Rating:** 7
**Confidence:** 4
**Correctness:** Yes
**Clarity:** The paper is well written.

**Strengths:**

- The proposed new tasks are different enough from the standard RL benchmarking tasks, e.g., delayed effects of action taken at a time, or drastic change in states due to an action.
- These tasks serve as a good starting point towards applying RL to real-world industrial manufacturing problems. The tasks are designed with safety in mind such as constraints on states.
- The experimental results are relatively complete as a large number of offline/online RL algorithms are benchmarked, which can serve as good reference for future benchmarkings on the proposed tasks.

**Weaknesses:**

- It would be good to have a discussion on how feasible the developed/trained RL algorithms on this benchmark can be appiled to the real-world applications. There are several abstractions in the tasks such as assuming access to all observable variables, ignoring sensor limiations such as noise.

**Additional Feedback:**

None

**Documentation:**

Yes

**Relation To Prior Work:**

Yes

**Summary And Contributions:**

This paper presents 5 novel industrial environments for benchmarking reinforcement learning algorithms. The tasks are inspired from real-world manufacturing process, have unique properties compared to traditional RL environments, and are well construcuted with expert domain knowledge. Some control baselines are performed on the proposed tasks and are used to collect datasets for offline RL algorithms. A suite of offline RL algorithms and a set of online RL algorithms are benchmarked on the proposed dataset, with some interesting conclusions/suggestions drawn from the experimental results.

---

> ### Author Response · Authors · 2022-08-13
> **Thanks for your feedback and we've added the Section 3.6: Limitations**
>
> **Weaknesses:**
>
> > It would be good to have a discussion on how feasible the developed/trained RL algorithms on this benchmark can be appiled to the real-world applications. There are several abstractions in the tasks such as assuming access to all observable variables, ignoring sensor limiations such as noise.
>
> Thanks for pointing this out! We have added the Section 3.6: Limitations in our paper: “Finally, while enabling machine learning practitioners and control engineers to explore more optimal control strategies and obtain actionable insights for real-world processes, the environments come with some limitations. Firstly, process variability and batch inconsistency is one of the major challenges in manufacturing, which cannot be simply described as gaussian noise. Adding more realistic noise and variability would make the simulations much closer to actual processes. Secondly, full accessibility to states may be difficult. For example, spectrophotometers are usually used for measuring chemical substances like concentrations, which could be costly. Last but not the least, state constraints are not fully considered. For example, due to safety and economic concerns, the next actions may not deviate too much from current actions. Although with the limitations described above, we believe that releasing the environments is still beneficial to researchers interested in applying reinforcement learning to control manufacturing processes. ” In conclusion, we cannot apply a working algorithm on a simulation directly to the factory. There should be human monitoring in the loop. The algorithm should suggest actions to take and experienced workers should decide whether to take the suggested actions or not.

---

### Official Review · Reviewer_5eM7 · 2022-07-31
**Review for SMPL**

**Rating:** 6
**Confidence:** 4

**Strengths:**

1. The major strength of the paper is that the authors introduce 5 new continuous control RL tasks to the community, which simulate important real-life manufacturing domains. I appreciate the great efforts put by the authors to develop the MDP formulations and make them realistic, which would include liaising with domain experts for building up the domains. Generally, I find the problems being studied rather interesting.
2. I feel it is very important to set up a solid benchmark standard for the following researchers to compare with and the paper introduces relatively extensive benchmark experiment settings. The authors consider employing not only offline and online RL algorithms, but also several related non-RL baselines, such the Bayesian Optimization. The derived benchmark scores are also very intuitive, reflecting the power of RL method as well as offline policy learning.
3. I like the statements presented by the authors on the difference between their proposed MDPs and the conventional ones studied by mainstream researchers in the community. The authors highlight some important properties of manufacturing control problems, such as delayed reward and drastic change in transition dynamics, which is inspiring for the RL researchers to develop new lines of RL problems.


**Weaknesses:**

1. The main limitation of this paper lies in the difficulty of understanding and evaluating the soundness of their proposed environments which seems rather complicated and requires extensive domain knowledge. The manufacturer problems are interesting ones and well motivated, but after reading the methodological part of it, the insights on how the MDPs are formulated for the problem, how the transition dynamics and reward functions are defined as well as how optimal policies should work is not shown. One suggestion is to move part of the differential functions into the main paper and explain them.
2. I’m also concerned about the simplicity of the problem. In real manufacturing domains, I feel the process of control is rather complicated, e.g., for beer fermentation, there are multiple control actions required to be issued at different time scales, but such problems might have been simplified in the MDP setting proposed for the domain by the authors.
3. The authors made several statements on the distinguishing properties of their proposed MDP compared to conventional RL domains  such as  the states considered SMPL are represented by flow-rates and volume of liquid which is very different from conventional RL domains, and the dynamics in SMPL change more dramatically than conventional domains. The drastic change of dynamics has not been proven by concrete examples, and I don’t feel SMPL’s state representation is that special compared to those in conventional MDPs.
4. In Table 1, the BeerFMTEnv is only evaluated by online algorithms, where the results for OfflineRL are missing. The authors assume the process for  beer fermentation can only be online in industrial production, but I think assumption might not hold as offline learning should also be appropriate as well like other domains.
5. The limitation of this work has not been discussed in detail. I am curious what are the limitations of employing the simulated environments to advise real-life manufacturing, and what are the difficulties of developing the online or offline training mechanism for manufacturing. For instance, to train the offline RL methods, the authors made an assumption on data generation  that the optimal trajectories are generated by some other algorithms, which might not be optimal.


**Additional Feedback:**

1. In Table 1, the authors highlight that the penalty for episodic failure should be less than $r_{min} * max_{stps}$, which is unclear. It is straightforward $e_r$ should be less than max episode reward. Do you mean the scale of the penalty should be less than cumulative reward instead? I also wonder how to practically setup a reasonable value for this failure in SMPL domains.
2. In Table 1, some domains request considerably smaller offline trajectories compared to others. What are the clues to specify the amount of offline data?


**Clarity:**

This paper is clearly written and easy to follow. However, the authors should improve the size and placement of each figure and table in the paper as well as the descriptions to them. I also feel the description to the MDP for each task is not complete, e.g. ,the detailed explanations to the action spaces and how the choice of actions affect transition dynamics and ultimate reward function presented in the main paper  is not thorough. There are also several typos and problems w.r.t the citations:
- Line 22 should be deleted.
- In Line 36, to align with -> and to align with
- In Line 61, multiple citations appeared for [6], and it’s better to specify corresponding ones for each problem domain, e.g., recommendation systems [14].
- In Line 73, inference of … is fast. -> the inference of … is faster.
- In Line 75, physics-simulation -> physics simulation
- In Line 97, the linebreak should be removed.
- In Line 98, PID has not been explained before mentioning it.
- Quite a few citations are incomplete with missing fields, e.g., [5][8][9][11][17][28][29][30][31][32][33][34][35][36][37][39][40][41][42][45][46][47][48][49][50][61].
- In Table 1, the entries are incomplete and some methods do not come with citations.


**Correctness:**

All the claims made in the submission are correct. The authors presented two concrete benchmarks (one for classification and one for AutoML) and clearly described the design criteria behind the benchmarks as well as examples of usage of both benchmarks.


**Documentation:**

The authors give sufficient detail and the URL to access the code and benchmarking suite.


**Ethics:**

This paper is related to empirical study on simulated domains with synthetic datasets. I feel there is no ethical concern about it.


**Relation To Prior Work:**

I feel the discussion on related works is not extensive. The authors focus on discussing the relationship between SMPL with Atari and Mujoco, but other established manufacturing MDP domains. I would expect to see more discussions on existing literatures on the specific perspective.


**Summary And Contributions:**

This paper proposes a new RL benchmark environment called SMPL, which consists of five Simulated Manufacturing Process Control tasks developed for important real-life manufacturing problems under the guidance from domain experts. The tasks being considered are: (1) beer fermentation; (2) a continuous stirred tank; (3) atropine production; (4) penicillin manufacturing; (5) monoclonal antibodies production. One thing needs to be highlighted is that the fifth task seems to be related to the COVID treatment, which is important to people’s lives today. The transitions in the MDPs are modeled by differential equations and the reward functions are safety constrained which promotes stability, less product investment and production yield.
The authors also present benchmark experiment results on their proposed domains. They consider several baselines tuned by domain experts, which include BO, PID, MPC and EMPC. They also develop several offline and online RL algorithms and testify them on the five domains. Benchmark results show that the RL methods could outperform non-RL baselines on AtropineEnv only.

---

> ### Author Response · Authors · 2022-08-13
> **Thanks for your feedback and we've revised the paper! To address the concerns... (1/3)**
>
> **Weaknesses:**
>
> > 1. The main limitation of this paper lies in the difficulty of understanding and evaluating the soundness of their proposed environments which seems rather complicated and requires extensive domain knowledge. The manufacturer problems are interesting ones and well motivated, but after reading the methodological part of it, the insights on how the MDPs are formulated for the problem, how the transition dynamics and reward functions are defined as well as how optimal policies should work is not shown. One suggestion is to move part of the differential functions into the main paper and explain them.
>
> Thank you for the feedback. We’ve gotten mixed reviews on this. On one hand, it is obviously helpful to understand the MDPs and transition dynamics; on the other hand, it is quite difficult to follow if the readers are not from the process control domain. With that in mind,  we tried to balance both sides. It is much appreciated if you could point out some of the details that you think should be included in the main paper and we will do our best to address them. Thanks again.
>
> > 2. I’m also concerned about the simplicity of the problem. In real manufacturing domains, I feel the process of control is rather complicated, e.g., for beer fermentation, there are multiple control actions required to be issued at different time scales, but such problems might have been simplified in the MDP setting proposed for the domain by the authors.
>
> Thank you for the feedback. To better address your comment, we have added limitations of the work to the paper in Section 3.6: “Finally, while enabling machine learning practitioners and control engineers to explore more optimal control strategies and obtain actionable insights for real-world processes, the environments come with some limitations. Firstly, process variability and batch inconsistency is one of the major challenges in manufacturing, which cannot be simply described as gaussian noise. Adding more realistic noise and variability would make the simulations much closer to actual processes. Secondly, full accessibility to states may be difficult. For example, spectrophotometers are usually used for measuring chemical substances like concentrations, which could be costly. Last but not the least, state constraints are not fully considered. For example, due to safety and economic concerns, the next actions may not deviate too much from current actions. Although with the limitations described above, we believe that releasing the environments is still beneficial to researchers interested in applying reinforcement learning to control manufacturing processes. ”

---

> > ### Author Response · Authors · 2022-08-13
> > **Continue… (2/3)**
> >
> > > 3. The authors made several statements on the distinguishing properties of their proposed MDP compared to conventional RL domains such as the states considered SMPL are represented by flow-rates and volume of liquid which is very different from conventional RL domains, and the dynamics in SMPL change more dramatically than conventional domains. The drastic change of dynamics has not been proven by concrete examples, and I don’t feel SMPL’s state representation is that special compared to those in conventional MDPs.
> >
> > Thank you for the comment. For “The drastic change of dynamics”, we wrote in line 71-73: “In chemical manufacturing, however, the pH (state) may not respond to a continuous flow of acid (action) after several minutes or even hours, but can also vary greatly with only a slight change in the concentration of such an input like in titration.”
> >
> > The [image](https://cdn.kastatic.org/ka-perseus-images/d77906ab32b311cc5d8025863233c34ed54c44eb.svg) from [Khan Academy Tutorial](https://www.khanacademy.org/test-prep/mcat/chemical-processes/titrations-and-solubility-equilibria/a/acid-base-titration-curves#:~:text=The%20point%20at%20which%20the%20indicator%20changes%20color%20is%20called,in%20an%20acid%2Dbase%20titration) is a good example: the pH of the analyte solution change drastically with a small volume of titrant added near the equivalence point. A brief explanation of titration would be like this: If we have an alkaline solution and we gradually add acid to the solution, the pH will barely change when the amount of acid is less than the base, and will change drastically when the amount of acid is equivalent or slightly more than the base. Titrant is a simple example of neutralization and the chemical process modeled in SMPL has many complicated reactions similar to titrant, as shown in Appendix A.
> > To summarize, the change of some of the states in SMPL can be both sharp and slow at different points than the change of states in games or kinematics. The pixels in a frame of a game will not likely be very different from the previous frame; the position of an end point of a robot will not likely to change suddenly in the next step. However, the pH value can change very slowly in one step, and then change sharply in another step (This usually occurs near the equivalence point).
> > As we described in lines 79 to 83, the dm_control simulations are specific and low-level physical simulations with accurate angles, coordinates or velocity. The kinematical abstraction makes the calculation of the simulation itself accurate. Even though we abstracted the manufacturing environments into ODEs, we should still notice the underlying difference in complexity between the simulation of a kinematic system and a manufacturing process. Therefore, one of the biggest differences between SMPL and canonical reinforcement environments is that, we should pay extra attention to the (unclear) distortions of the abstracted ODEs approximations and constraint ourselves harshly. Moreover, because of the complexity of the chemical and biological reactions, it would be hard for us to give clear constraints. We sometimes penalize the deviation from a steady state or canonical actions, but the development of those constraints may still be further improved despite our effort given in the environments and the reward functions. We’ve included some of the above discussion in section 3.6.
> >
> > > 4. In Table 1, the BeerFMTEnv is only evaluated by online algorithms, where the results for OfflineRL are missing. The authors assume the process for beer fermentation can only be online in industrial production, but I think the assumption might not hold as offline learning should also be appropriate as well like other domains.
> >
> > Thanks for the feedback. We didn’t assume BeerFMTEnv can only be online, we didn’t perform the offline RL experiments due to the limitation of the provided recipes. As described in lines 174 and 175, “Since we just have a canonical industrial production formula that can only solve from a fixed initial state, we would only perform online reinforcement learning experiments.”

---

> > > ### Author Response · Authors · 2022-08-13
> > > **Continue… (3/3)**
> > >
> > > > 5. The limitation of this work has not been discussed in detail. I am curious what are the limitations of employing the simulated environments to advise real-life manufacturing, and what are the difficulties of developing the online or offline training mechanism for manufacturing. For instance, to train the offline RL methods, the authors made an assumption on data generation that the optimal trajectories are generated by some other algorithms, which might not be optimal.
> > >
> > > Thanks for the feedback. The offline datasets are generated with hand-tuned MPC/BO algorithms. as written in section 5.1, “ Each of the four simulation environments has an expert control algorithm which is manually tuned by experts. This expert control algorithm can provide successful controls in non-extreme cases, with potentially low efficiency due to online optimization and computation complexity.” However, their tuning is not optimal. The MPC, for example, can be sub-optimal or even break the simulation with unsafe behaviors for many initial states in ReactorEnv, AtropineEnv and mAbEnv, resulting in error_reward to be returned. A good offline RL algorithm should learn from the generated offline datasets how MPC fails and succeeds, then outperform the MPC in the simulation environments. We discussed the tested Offline RL algorithms in section 5.1.
> > > For limitations, we have added a new section 3.6, as quoted in the above reply.
> > >
> > > **Clarity:**
> > >
> > > > This paper is clearly written and easy to follow. However, the authors should improve the size and placement of each figure and table in the paper as well as the descriptions to them. I also feel the description to the MDP for each task is not complete, e.g. ,the detailed explanations to the action spaces and how the choice of actions affect transition dynamics and ultimate reward function presented in the main paper is not thorough. There are also several typos and problems w.r.t the citations:
> > >
> > > Thanks for the feedback. Due to the limitation of the preprint version, we might not be able to have the best placement of figures now. We’ve adjusted the size of some figures and we will re-adjust the figures and tables in the camera-ready version once the length of the contents is settled.
> > > For the MDP, due to the space limitation, we were not able to fully discuss them. We included the details of transition dynamics, action space, state space and reward function in the appendices. Readers interested in technical details could also read the documentation https://smpl-env.github.io/smpl-document/index.html and the code https://github.com/smpl-env/smpl/tree/main/smpl/envs.
> > >
> > > > Line 22 should be deleted.
> > >
> > > Thank you for the suggestion. We have deleted it.
> > >
> > > --In Line 36, to align with -> and to align with
> > >
> > > We agree it sounds misleading. We changed it to:
> > > “With the help of domain experts, we managed to access and utilize the industrial data to validate our models and determine the corresponding parameters, to align with real-world factories.” in line 38.
> > >
> > > --In Line 61, multiple citations appeared for [6], and it’s better to specify corresponding ones for each problem domain, e.g., recommendation systems [14].
> > >
> > > Thanks for pointing this out. We’ve changed accordingly in line 62.
> > >
> > > > In Line 73, inference of … is fast. -> the inference of … is faster.
> > >
> > > Thank you for pointing out the typo to us. We have corrected and it now reads:
> > > “inference of deep reinforcement learning algorithms is faster.” in line 76.
> > >
> > > > In Line 75, physics-simulation -> physics simulation
> > >
> > > Thank you for pointing out the typo to us. We have corrected and it now reads:
> > > “other MuJoCo [18] based physics simulation environments” in line 78
> > >
> > > > In Line 97, the linebreak should be removed.
> > >
> > > Thank you for the suggestion. We have removed the linebreak and it now reads:
> > > “when the agent is exploring outside the reliable region. Therefore, compared to the results of reinforcement learning experiments” in line 98.
> > >
> > > > In Line 98, PID has not been explained before mentioning it.
> > >
> > > We’ve changed “PID” to “Proportional–Integral–Derivative (PID) controller” in line 100. Thanks for pointing this out!
> > >
> > > > Quite a few citations are incomplete with missing fields, e.g.,
> > > [5] [8] [9] [11] [17] [28] [29] [30] [31] [32] [33] [34] [35] [36] [37] [39] [40] [41] [42] [45] [46] [47] [48] [49] [50] [61]
> > >
> > > Thank you for the comment. We have carefully gone through the bibliography and corrected the incomplete citations.
> > > > In Table 1, the entries are incomplete and some methods do not come with citations.
> > >
> > > Thank you for the comment. The incomplete entries are experiments that are not performed or not available. We’ve also added the hyperlinks for the methods that we introduced in Section 4 Baseline Algorithms.

---

### Author Response · Authors · 2022-08-25
**Your Engagement Would Be Much Appreciated!**

Thank you to all the reviewers for acknowledging the novelty of the benchmark in introducing manufacturing control problems into the reinforcement learning domain.

Multiple reviewers agreed that we provided a thorough description of simulation environments and that improvements in applications of learning-based methods in this domain would be of broad interest. There is also detailed documentation and explanation of the environments alongside all the code published, to enable turn-key reproduction of the results in the paper.

The reviewers also provided helpful critiques on how to improve the paper.
We have replied to all the concerns and addressed them accordingly by revising the paper.

One concern was the level of detail about the environments. Due to the limitation of space and the concern of readability, the chemical and biological environments are discussed briefly in the main paper, while the technical details are covered in the Appendices and the documentation. We have however added relatively concise explanations to address the concerns of [Reviewer *5eM7*](https://openreview.net/forum?id=TscdNx8udf5&noteId=fsYXQ7K-QR0).
We also addressed possible confusion and added more structured details in our [documentation](https://smpl-env.github.io/smpl-document/api/smpl.envs.html).

Furthermore, we added a new section **3.6 Limitations** to discuss the limitations of our environment and why it could potentially be hard to directly apply working reinforcement learning algorithms to real producing factories.
We have also added new text (the last paragraph) in the **Related Work** section to discuss the difference between our work and some related works, to show that the major contribution of our work is to set up a solid, clear and easy-to-follow benchmark standard for all the following reinforcement learning research in the manufacturing control area.

Thank you again for all your time and effort in reviewing our paper. It would be much appreciated if you could let us know whether we have addressed all the concerns and how we could provide further clarifications or improve our paper.

---

### Meta-Review · Area_Chair_8Szw · 2022-09-02

**Recommendation:** Accept
**Confidence:** 4

**Metareview:**

The paper provides a suite of environment simulators aimed for offline and online RL evaluation. The contributions of the paper are significant for focusing on industrial manufacturing and process control problems, which is a departure from more classic RL tasks (e.g., locomotion, robotic manipulation, driving). These environments provide a diverse set of compelling applications that I hope will help to widen the impact of RL research. Overall, the reviewers are positive, and I was unable to get a response from the only remaining negative reviewer (RKJs) regarding whether their concerns were addressed, and so I lean towards accepting the paper. I do want to note to the authors that a lingering concern shared by me and the reviewers is, how realistic these simulators actually are? It would help if the authors could be more explicit in the paper for each environment, whether it is an exact (or near-exact) analogue of what practitioners use for testing, and if not, what are the main simplifications?

---

### Decision · Program_Chairs · 2022-09-16

Accept